# An Update on Self-Amplifying mRNA Vaccine Development

**DOI:** 10.3390/vaccines9020097

**Published:** 2021-01-28

**Authors:** Anna K. Blakney, Shell Ip, Andrew J. Geall

**Affiliations:** 1Michael Smith Laboratories, School of Biomedical Engineering, University of British Columbia, Vancouver, BC V6T 1Z4, Canada; 2Precision NanoSystems Inc., Vancouver, BC V6P 6T7, Canada; sip@precision-nano.com (S.I.); ageall@precision-nano.com (A.J.G.)

**Keywords:** RNA, self-amplifying RNA, replicon, vaccine, drug delivery

## Abstract

This review will explore the four major pillars required for design and development of an saRNA vaccine: Antigen design, vector design, non-viral delivery systems, and manufacturing (both saRNA and lipid nanoparticles (LNP)). We report on the major innovations, preclinical and clinical data reported in the last five years and will discuss future prospects.

## 1. Introduction: The Four Pillars of saRNA Vaccines

In December 2019, the SARS-CoV-2 (severe acute respiratory syndrome coronavirus 2) virus emerged, causing a respiratory illness, coronavirus disease 2019 (COVID-19), in Hubei province, China [1,2]. The virus has spread globally, with the World Health Organization (WHO) declaring it a Public Health Emergency of International concern on 30 January 2020 and a pandemic officially on 7 March 2020 [3]. There is a strong consensus globally that a COVID-19 vaccine is likely the most effective approach to sustainably controlling the COVID-19 pandemic [4]. There has been an unprecedented research effort and global coordination which has resulted in the rapid development of vaccine candidates and initiation of human clinical trials. This has included conventional vaccine technologies such as viral vectors and adjuvanted subunits, but we have witnessed a renaissance in the field of RNA vaccines and a shift towards synthetic RNA platforms (Figure 1) [5,6]. In fact, one of the first vaccine to start clinical trials was a non-replicating mRNA vaccine from Moderna, mRNA-1273 [7,8,9]; the first patient was vaccinated on 16 March at the same time as a Chinese clinical trial was initiated with an adenovirus type-5 (Ad5) vector [10]. Furthermore, the BioNTech/Pfizer vaccine, BNT162b2, was the first COVID-19 vaccine to receive approval, first in the United Kingdom and then Canada, with an impressive 95% efficacy [11]. Since this time there have been several mRNA vaccine trials initiated, and publication of corresponding preclinical and clinical data, see Table 1.

The use of mRNA vaccines for pandemic response has been well described previously in preclinical [12,21,22,23,24,25,26,27,28,29,30,31,32,33] and clinical settings [25], but this is the first time we have seen the platforms deployed in a real pandemic setting [34]. The core principle behind mRNA vaccines is to encode the antigen in the mRNA and then to deliver the transcript to the host cell cytoplasm using a non-viral delivery system, allowing antigen expression and induction of an antigen-specific immune response. This is especially advantageous as a vaccine platform as mRNA vaccines can be produced for any pathogen with a known protein target. mRNA is made using a cell-free enzymatic transcription reaction, which allows rapid and scalable manufacturing, as is evident from the swift pursuit of RNA vaccines in the current pandemic. Currently, there are three major types of RNA vaccines: conventional, non-amplifying mRNA molecules (mRNA), base-modified, non-amplifying mRNA molecules (bmRNA), which incorporate chemically modified nucleotides, and self-amplifying mRNA (saRNA or replicons) that maintain auto-replicative activity derived from an RNA virus vector. Self-amplifying RNA is beneficial compared to non-amplifying RNA as it maintains the advantages of mRNA vaccines, such as rapid development, modular design, and cell-free synthesis, but requires a lower dose of RNA due to the self-replicative properties. This reduces the burden of manufacturing for both the drug substance and product and is potentially advantageous in the context of pandemic response as it would enable a greater percentage of the population to be vaccinated in a shorter amount of time.

This review will explore the four major pillars required for design and development of an saRNA vaccine (Figure 2): Antigen design, vector design, non-viral delivery systems, and manufacturing (both saRNA and lipid nanoparticles (LNP)). In will report on the major innovations, preclinical and clinical data reported in the last five years and will discuss future prospects (Figure 3). Pertinent reviews on plasmid DNA and non-replicating messenger RNA vaccines can be found at the following references, which provide insight into the mechanism of immune response and effects of route of delivery [35,36,37,38].

## 2. Antigen Design

saRNA vaccines have been primarily investigated for active vaccination strategies for prevention of infectious diseases, wherein the host’s cells produce a pathogenic antigen encoded in saRNA to induce a humoral and cellular immune response. saRNA encoding viral glycoproteins are the most predominant application, although this has recently been expanded to include bacterial infections (*Chlamydia trachomatis* [57], Group A and B *Streptococci* [58]), parasites (*Toxoplasma gondii* [59,60]) and cancer (colon carcinoma, [61,62] melanoma [62]). A more novel approach to saRNA antigen design includes encoding monoclonal antibodies for passive vaccination [63]. While it is possible to incorporate relatively large (>4000 nt) or multiple antigens into an saRNA construct, the pDNA construct does have size limitations, so it may be advantageous to use separate saRNA constructs to encode multiple antigens if necessary [64].

### 2.1. Infectious Diseases

#### 2.1.1. Viral Glycoproteins

Recent advances in saRNA vaccines against infectious diseases include development of vaccines against a variety of viral pathogens. The breadth of these vaccines includes respiratory-transmitted viruses (SARS-CoV-2, respiratory syncytial virus, influenza), insect-transmitted viruses (VEEV, Zika, Ebola), animal-transmitted viruses (rabies), and sexually transmitted viruses (HIV-1) (Table 2). Samsa et al. observed that a codon-modified VEEV backbone, with the positively charged amino acid residues at the N-terminal region of the capsid protein (CP) mutated to non-charged residues, induced lower IgG and neutralization titers compared to the wild type, although these mutations had been previously observed to increase VEEV replication [65]. Importantly, Magini et al. showed that it’s possible to co-deliver saRNA encoding multiple antigens, in this case the influenza nuclear and M1 proteins, to induce heterospecific neutralizing antibodies that protect against heterologous challenge [66]. Furthermore, all the clinical trials currently underway for saRNA vaccines have viral glycoproteins as the target (Table 3).

#### 2.1.2. Bacterial Antigens

saRNA vaccines against bacterial antigens have also been investigated, although are limited to protein targets, as opposed to polysaccharides and non-protein surface markers. Maruggi et al. investigated the immunogenicity and efficacy of saRNA against Group A and Group B *Streptococci*, as model bacterial pathogens [58]. They used saRNA encoding Streptolysin-O (SLOdm) and pilus 2a backbone protein (BP-2a), and achieved partial protection against intraperitoneal infection in a maternal immunization/pup challenge model, although protection was higher in both cases with the recombinant protein vaccine. Blakney et al. investigated the use of saRNA encoding the major outer membrane protein (MOMP) of *Chlamydia trachomatis* as a model antigen, complexed with cationic adjuvant formulations (CAFs) [57]. The three saRNA formulations all exhibited antigen-specific humoral and cellular immunity against MOMP, although a challenge study was not included as part of this study. These studies show that it is possible to use saRNA vaccines against bacterial pathogens as a means of disease prevention.

#### 2.1.3. Parasitic Antigens

saRNA vaccines have been applied to two parasitic indications, *Toxoplasma gondii* and *Plasmodium*. Chahal et al. demonstrated that a hexaplex saRNA vaccine with six *T. gondii*-specific antigens, including dense granule protein 6 (GRA6), rhoptry protein 2A (ROP2A), rhoptry protein 18 (ROP18), surface antigen 1 (SAG1), surface antigen 2A (SAG2A), and apical membrane antigen 1 (AMA1), protected mice against lethal *T. gondii* challenge with a dose of 6.67 µg per replicon (40 µg total) after a single IM injection [59]. Luo et al. utilized saRNA encoding nucleoside triphosphate hydrolase-II (NTPase-II) with a prime-boost regimen of 10 µg doses and observed partial protection, prolonged survival time and reduction in brain parasitic load [60]. Baeza Garcia et al. vaccinated mice with a replicon encoding *Plasmodium* macrophage migration inhibitory factor (PMIF) and showed that the vaccine delayed blood-stage patency after sporozoite infection, increased anti-*Plasmodium* antibody titers, and completely protected from reinfection [73]. These studies demonstrate that the saRNA platform can also prevent parasitic infection.

#### 2.1.4. Monoclonal Antibodies for Passive Vaccination

As opposed to activate vaccination, passive vaccination with monoclonal antibodies (mAb) provides more immediate protection against a pathogen. Because monoclonal antibodies are expensive to produce and difficult to administer, mRNA is a highly useful alternative platform. Previous studies have utilized mRNA [95] or pDNA [95] encoding a neutralizing antibody against chikungunya virus or influenza and Ebola viruses, respectively. Though the mRNA LNP formulation is protective against chikungunya virus challenge, the required doses of 40–200 µg in mice preclude application of this technology in a human trial. Erasmus et al. encoded ZIKV-117, a potent neutralizing mAb, and observed that a 40 µg dose of saRNA induced higher levels of systemic antibody than an equivalent dose of mRNA. While the circulating antibody levels were protective against Zika virus challenge, the titers reached a maximum concentration of 2 µg/mL, which could likely be improved with molecular and delivery platform optimization. While this strategy is advantageous for infectious diseases with a known neutralizing antibody, it may also be applied to mAb treatments against cancers or rare and inherited diseases.

### 2.2. Cancer

While mRNA vaccines have been widely applied to oncology, [96,97] including for the generation of neoantigen cancer vaccines, recent advances in saRNA cancer vaccines are more limited. Li et al. used a clever in vitro evolution approach to introduce mutations into the VEEV replicon backbone that enhanced the magnitude and duration of protein expression in vivo [62]. Compared to the wild-type replicon, the evolved saRNA showed a 5.5-fold increase in the intra-tumoral interleukin-2 (IL-2) levels and increased infiltrating CD8^+^ T-cells, which resulted in significantly slowed melanoma tumor growth. Li et al. also developed an LNP-formulated saRNA encoding IL-12 to stimulate immunogenic cancer cell death (ICD) by utilizing an LNP composition that itself stimulates ICD, saRNA that triggers cellular activation, and interleukin-12 (IL-12) for immunomodulation. The observed that the saRNA LNPs induced a highly inflamed tumor microenvironment, eradicated large established tumors and regression of distal un-injected tumors. Gritstone Oncology, Inc. has two ongoing clinical trials using saRNA personalized neoantigen vaccines against non-small cell lung cancer, colorectal cancer, gastroesophageal adenocarcinoma, urothelial carcinoma, solid tumors, and pancreatic ductal adenocarcinoma (Table 3), although preclinical studies and trials results have not yet been published. Virus replicon particles (VRP) have been utilized more extensively clinically in the cancer vaccine space, with encouraging data presented in these reviews [30,98]; these strategies will likely transition to non-viral delivery approaches in the future. Together these studies set a precedent for future use of saRNA vaccines for cancer applications.

## 3. Vector Design

As discussed in the introduction, there are three major forms of RNA vaccines based on the auto-replicative capacity of the mRNA and the inclusion of mammalian base-modifications. This section will focus on saRNA, or replicons, that maintain replicative activity derived from an RNA viral vector. Historically, positive-sense single-stranded RNA viruses, such as alphaviruses, flaviviruses, and picornaviruses have been used for replicons. The best-studied self-amplifying mRNA molecules are derived from alphavirus genomes, such as those of the Sindbis virus, which have been previously reviewed in references [5,24,30,99,100]. This section explores how saRNA self-amplifies and any new published insights that might help in the rational design of vectors. In addition, it highlights any innovations reported in the last 5 years on the design of saRNA vectors.

### 3.1. Mechanisms of Self-Amplification of RNA

saRNAs are considerably larger (≈9–12 kb) than non-amplifying mRNAs (Figure 4). They contain the basic elements of mRNA (a cap, 5′ UTR, 3′ UTR, and poly(A) tail of variable length) but have a large open reading frame (ORF) at the 5′ end that encodes four non-structural proteins (nsP1–4) and a subgenomic promoter. Genes in the viral genome that are normally downstream of the subgenomic promoter and encode the viral structural proteins are replaced by gene(s) encoding the vaccine antigen(s). Deletion of the viral structural proteins renders the mRNA incapable of producing an infectious virus. After delivery into the cytosol of a cell, the released mRNA is translationally competent, and engages with the host cell ribosome to produce the four functional components of RNA-dependent RNA polymerase (RDRP) or viral genome replication apparatus: nsP1, nsP2, nsP3 and nsP4 (Figure 5). Studies on the regulation of alphavirus RNA synthesis, the roles of the viral non-structural proteins in this process and the functions of *cis*-acting RNA elements in replication have led to a greater understanding, but there are still knowledge gaps that restrict rational design of new vectors [101]. Formation of the RDRP is a complex, multistage process, with each of the nsPs having several functions [102,103,104]. These proteins are expressed as a polyprotein and processed in a highly regulated manner into individual proteins by the viral protease (nsP2). nsP1 is required for plasma membrane association of the replicase complex and 5′ capping of viral RNA species while nsP2 serves as RNA helicase and protease for polyprotein processing, nsP3 exerts a crucial function in mediating multiple virus–host protein–protein interactions, and nsP4 is the RNA-dependent RNA polymerase. Viral RNA synthesis requires the appropriate recognition of sequence and structural elements in the template RNAs by the viral RNA synthetic complex [101]. For alphaviruses, *cis*-acting elements predominantly correspond to UTRs, of which there are three, located at the 5′ end, the 3′ end, and the junction region between the non-structural and structural ORFs. These UTRs have functions and new research is starting to provide greater insights [105]. In addition, elements exist in coding regions of the genome and subgenome that function in the synthesis of viral RNA, viral protein expression and viral genome packaging. These elements are conserved to varying degrees across the genus, and their role(s) in alphavirus replication continues to be clarified and refined [105,106,107].

The RDRP complex is tethered to the plasma membrane (PM) in a bulb-shaped membrane invagination, where it is hidden from host cell immune surveillance [103]. The viral replicase first uses the positive sense genome as template to synthesize complementary negative sense RNA which subsequently serves as template for the synthesis of genomic and subgenomic plus-strand RNA. The subgenomic RNA is produced in excess of the viral genome [24]. This process leads to high and sustained levels of antigen expression relative to conventional mRNA and is certainly one of the reasons saRNA vaccines require lower doses of RNA [78]. RNA self-amplification in transfected cells also leads to cellular exhaustion, immune stimulation through dsRNA intermediates and a host cell antiviral response leading to apoptosis. In many ways, this process mimics a viral infection and leads to enhance antigen-specific B and T cell responses [75,88]. In parallel to the self-amplification process, which occurs primarily in myocytes at the site of intramuscular vaccination [75], the input saRNA leads to stimulation of the innate immune system. This sensing is mediated by pattern-recognition receptors (PRRs), which detect conserved pathogen-associated molecular patterns (PAMPs) on the nucleic acid [108]. Detection of PAMPs by PRRs leads to the induction of inflammatory responses and innate host defenses. In addition, the sensing of saRNA by PRRs expressed by antigen-presenting cells, particularly dendritic cells (DCs), leads to the activation of adaptive immune responses [108]. Over the last 5 years, saRNA vaccine mechanism of action studies and a better understanding of the RNA amplification process have led to new areas of vector innovation [88,109].

### 3.2. Innovative Self-Amplifying RNA Vector Designs

In the last five years, there have been progressive designs of RNA replicons to introduce superior mutations and pioneer the use of *trans*-amplifying RNA systems. Li et al. developed an in vitro evolution strategy and identified six mutations in nonstructural proteins (nsPs) of Venezuelan equine encephalitis (VEE) replicon that promoted subgenome expression in cells [62]. Furthermore, a research team at Imperial College London developed a split replicon (splitzicon) system wherein the non-structural proteins (NSPs) and the gene of interest are encoded on separate RNA molecules, but still exhibit the self-amplification properties of replicon RNA [110]. They designed both positive and negative strand splitzicons encoding firefly luciferase as a reporter protein to determine which structural components affect amplification. In vitro proof of concept was demonstrated, highlighting a system for screening the components required for amplification from the positive and negative strand intermediates of RNA replicons that might lead to future vector improvements. Subsequent to this work, Beissert et al. have developed a novel bipartite vector system using *trans*-amplifying RNA (taRNA) [82]. The vector cassette encoding the vaccine antigen originates from an alphaviral self-amplifying RNA (saRNA), from which the replicase was deleted to form a *trans*-replicon. Replicase activity is provided in *trans* by a second molecule, an optimized non-replicating mRNA (nrRNA). Expression driven by the nrRNA-encoded replicase in the taRNA system was as efficient as a conventional monopartite saRNA system in a mouse influenza challenge model [82].

### 3.3. Improving Immunogenicity with Molecular Interferon Modulators

It is well known that saRNA activates the type I interferon (IFN) response through both endosomal sensing, via toll-like receptor (TLR) 3, 7 and 8, and cytosolic sensing via melanoma differentiation-associated protein 5 (MDA5), retinoic acid-inducible gene I (RIG-I), protein kinase R (PKR), 2′-5′oligoadenylate synthetase (OAS) as well as other possibly unknown pathways [38,111]. While this is advantageous for enhancing the immunogenicity of saRNA vaccines, IFN activation is also known to lead to inhibition of translation [112] and degradation of cellular mRNA [113], which may hinder the potency of the vaccine. To counter IFN activation, the co-delivery of viral immune evasion proteins (E3, K3, and B18/B18R from vaccinia virus and nonstructural protein 1 (NS1) from flu) are being explored to reduce immune signaling and have shown potential [5,114,115,116]. Beissert et al. co-delivered non-replicating mRNA encoding vaccinia virus immune evasion proteins E3, K3 and B18 with saRNA [114]. They observed that co-delivery of the E3 protein, which counteracts translation arrest by ensuring eukaryotictranslation initiation factor 2 α (eIF2α) functionality, enhanced saRNA expression both in vitro and in vivo. The downfall of this approach is that the 2 µg dose of saRNA required co-delivery of either 6 or 12 µg of E3 mRNA, which significantly increases the amount of administered RNA. Furthermore, *trans*-encoding these proteins may limit the number of cells that take up and express both types of RNA. Blakney et al. improved upon this approach by encoding an interferon inhibiting protein (IIP), screened from a library of known viral immune evasion proteins, directly in the saRNA [64]. They observed that the parainfluenza virus 5 (PIV-5) V protein and the Middle East Respiratory Syndrome (MERS) ORF4a protein enhanced protein expression both in vitro and in vivo in mice, and immunogenicity of saRNA encoding the rabies G glycoprotein in rabbits. Interestingly, they also observed that ruxolitinib, a janus kinase (JAK)/ signal transducer and activator of transcription (STAT) inhibitor [117], increased protein expression in vivo, but did not test the effects on immunogenicity. These approaches provide proof-of-concept that saRNA expression and immunogenicity can be favorably impacted by expression of interferon inhibiting proteins.

Overall, VEEV and SINV vector designs have been shown to have the most promising vaccine immunogenicity and are being improved by next generation designs such as *trans*-amplifying replicons and incorporation of interferon inhibiting factors directly into the vector backbone.

## 4. Delivery Systems

The main challenge for saRNA vaccines is achieving sufficient delivery of saRNA to the target cells or tissue. saRNA constructs are relatively large (9000 to 15,000 nt), anionic molecules, which precludes efficient cellular uptake of unformulated saRNA. Despite the use of “naked” saRNA in some studies, three predominant delivery platforms have emerged: Polymeric nanoparticles, lipid nanoparticles, and nanoemulsions. These delivery strategies share a central dogma wherein the anionic saRNA is condensed by a cationic (or ionizable cationic) carrier to a nanoparticle of ~100 nm in size, that protects the saRNA from degradation and encourages uptake into target cells (Figure 6). Relevant studies with recent advances (since 2015) using saRNA vaccines can be found in Table 2.

### 4.1. Naked saRNA

Naked saRNA has been successfully used for in vivo immunizations against HIV-1 subtype C [72], influenza [82], and Zika viruses [92]. While these studies observed that the naked saRNA induced humoral and/or cellular responses, the required dose was significantly higher than other saRNA vaccine studies, and similar to doses used for mRNA. Abjani et al. observed *Env*-specific antibodies and induction of *gag*-specific IFN-y secreting splenocytes after three intramuscular immunizations of 20 µg of saRNA [72]. Similarly, Beissert et al. immunized mice intradermally against H1N1 influenza using a *trans*-amplifying replicon system comprised of 20 µg of the replicase and varying doses (0.05 to 31.25 µg) of the hemagglutinin (HA) antigen, and observed complete protection of mice against influenza challenge [82]. Zhong et al. utilized electroporation to deliver a dose of 1 or 10 µg of saRNA intramuscularly and observed moderate antibody and cellular responses against the precursor membrane (PrM) and envelope (E) proteins of Zika virus [92]. These studies demonstrate that while it is possible to induce immune responses using naked saRNA, the dose required eliminates any advantage of using saRNA over non-replicating mRNA. Interestingly, Huysmans et al. observed that electroporating saRNA significantly enhanced the expression kinetics compared to naked or LNP-formulated saRNA, which they postulate was due to a limited innate immune response after intradermal injection [118]. This important finding highlights that the innate response to the saRNA delivery platform can profoundly impact immunogenicity.

### 4.2. Polymeric Nanoparticles

Polymeric nanoparticle delivery platforms for saRNA can segregated into non-degradable and degradable polymers. Polyethyleneimine (PEI) is a non-degradable, cationic polymer that has been used by a number of groups for delivery of saRNA. Vogel et al. demonstrated that PEI-formulated saRNA protected against three strains of influenza (H1N1, H3N2, and B), and required a 64-fold lower dose compared to mRNA [78]. Démoulins et al. observed that linear PEI induced humoral and cellular immune responses against influenza HA and NP through efficient internalization in dendritic cells (DC) [79]. Following on this work, Démoulins et al. demonstrated that increasing the molecular weight (MW) of PEI inhibits internalization of polyplexes, but that adding Arg_9_, a cell penetrating peptide (CPP), modestly enhanced cellular responses to PEI-formulated saRNA in pigs [76]. Chahal et al. aimed to improve saRNA polyamine delivery by utilizing monodisperse, molecular defined dendrimers, and showed induction of protective immunity against influenza, Ebola and *Toxoplasma gondii* challenges using modified dendrimer nanoparticles (MDNP) [90]. Because PEI is known to be cytotoxic, especially at higher molecular weights [119], but higher MW PEI-based polymers enhanced the transfection efficiency of saRNA [120], Blakney et al. developed pABOL, a bioreducible, cationic polymer which was shown to enhance transfection efficiency, but not cytotoxicity, at higher MW and to protect mice from influenza challenge at a dose as low as 1 µg [83].

While the ideal target cells for saRNA vaccines are not yet defined, recent polymeric nanoparticles have been developed to target saRNA polyplexes to different cell populations. Gurnani et al. observed that increasing the hydrophobicity of poly(dimethylaminoethyl) acrylate (pDMAEA) copolymers enhances saRNA expression in epithelial cells in human skin explants after intradermal injection [121]. Blakney et al. observed that mannosylated-PEI polyplexes similarly enhanced saRNA expression in epithelial cells in human skin explants in a mannose-dependent manner [122]. Saviano et al. showed that increasing the branching of orthenine-derived dendrimers enriched saRNA uptake and expression specifically in epithelial, NK and Langerhans cells [123]. Ultimately, these targeting strategies may enable targeted delivery of saRNA vaccines to enhance efficiency.

### 4.3. Lipid Nanoparticles

Lipid nanoparticle formulations of saRNA are currently the most potent, requiring as little as 10 ng of saRNA to induce a robust immune response [18]. saRNA LNPs are predominantly based on formulations optimized for siRNA and mRNA delivery, which include an ionizable lipid, phospholipid, cholesterol, and PEGylated lipid [52]. These LNPs have been used for a variety of saRNA vaccine infectious disease indications, including SARS-CoV-2 [18], influenza [66], rabies [87], *Toxoplasma gondii* [60], respiratory syncytial virus [88], as well as recent advances in saRNA cancer vaccines, including melanoma [62] and colon carcinoma [61]. Melo et al. used LNPs based on the cationic lipid 1,2-dioleoyl-3-timethylammonium-propane (DOTAP), and showed high titers of gp120-specific antibodies after a single intramuscular injection, as well as increased levels of antigen-specific germinal center B cells compared to protein immunization [70].

While saRNA has historically been encapsulated on the interior of lipid nanoparticles, there have also been recent advances of LNP formulations wherein the lipid particle is formed and then the saRNA is complexed on the surface of the particle. Blakney et al. showed induction of HIV-1 gp140 antibody responses was higher with cationic-based lipoplexes, although protein expression was highest when saRNA was encapsulated within an ionizable LNP [71]. Furthermore, Blakney et al. observed that lipoplexes prepared with dimethyldioctadecylammonium (DDA) induced humoral and cellular immune responses against *Chlamydia trachomatis* [57]. Interestingly, Englezou et al. demonstrated that it was possible to deliver saRNA and induce immune responses against influenza by simply complexing the saRNA with DOGTOR, a cationic lipid [77]. These studies demonstrate the versatility and potency of the lipid-based delivery platforms.

### 4.4. Nanoemulsions

Cationic nanoemulsions (CNE) are also a leading strategy for delivery of saRNA vaccines. The emulsions are typically a water-in-oil emulsion, similar to the license MF59 adjuvant, that consists of squalene, sorbitan trioleate, polysorbate 80 and DOTAP [124]. The main advantage of this platform is that MF59 has a well-defined safety profile in humans [125]. Anderluzzi et al. observed that CNE had the highest induction of antibodies against rabies in a direct comparison with DOTAP polymeric nanoparticles, DOTAP liposomes and DDA liposomes [86]. Bogers et al. showed in the first in nonhuman primate study that CNE enabled immunogenicity equivalent to an adjuvanted protein vaccine against a clade C glycoprotein of HIV-1 [68]. CNE is also a versatile delivery platform, and has been shown to generate immune responses against a variety of pathogens including Group A and B *Streptococci* [58], HIV-1 [68], influenza [74], rabies [86], and VEEV [65]. These studies show that nanoemulsions are a potent and versatile delivery platform for saRNA vaccines.

### 4.5. Adjuvanted Delivery Systems

saRNA is considered to be self-adjuvanting due to the dsRNA structures, replicon intermediates and other motifs that are sensed intracellularly [38]. However, recent studies have investigated the role of both the delivery vehicle and molecular components in adjuvanting saRNA vaccines. Blakney et al. observed that the adjuvancy of incorporating 3M-052, a TLR 7/8 agonist, into lipoplexes was eclipsed by the self-adjuvanting effects of saRNA. Démoulins et al. found that incorporating Pam3Cys-SK4 (P3C), a bacterial lipoprotein, promoted saRNA internalization by DCs in vitro but did not enhance humoral or cellular immunogenicity in vivo [79]. Manara et al. found that encoding granulocyte-macrophage colony-stimulating factor (GM-CSF), a chemoattractant, directly in saRNA increased the recruitment of antigen presenting cells (APCs) to the site of injection and increased antigen-specific CD8^+^ T-cell responses, but did not affect humoral immunity [84]. These studies give insight into strategies regarding enhancing the immunogenicity saRNA vaccines using either molecular or biomaterial adjuvants.

### 4.6. Delivery Platforms in the Clinic

The momentum of the field of RNA gene delivery has accelerated in recent years given the 2018 FDA approval of the LNP-formulated siRNA drug, Onpattro [126], and the recent shot of adrenaline to RNA vaccines in general provided by the COVID-19 global pandemic. There are currently three ongoing saRNA vaccine clinical trials and two in the pre-recruiting phase (Table 3), all of which use either LNP or CNE as a delivery platform. GSK is currently evaluating a VEE-SINV chimeric replicon encoding the rabies glycoprotein formulated with CNE at three different doses; this study is in Phase I and is estimated to complete in April 2021. The Shattock laboratory at Imperial College London is evaluating a VEEV replicon encoding the pre-fusion stabilized spike protein of SARS-CoV-2 formulated in LNP at doses ranging from 0.1–10 µg; this study is in Phase II of a combined Phase I/II trial and is estimated to complete in July 2021. Finally, Arcturus Therapeutics is also testing a saRNA vaccine encoding the prefusion spike protein of SARS-CoV-2 formulated in LNP at four doses and is in Phase II of combined Phase I/II trial slated to complete in December 2020. Two upcoming clinical trials will test a VEEV saRNA vaccine given IN directly against ChAdOx [127] and a saRNA VEEV vaccine formulated with CNE against SARS-CoV-2. The major considerations for these clinical trials, other than the humoral and cellular immunity, are the required dose, the vaccine schedule and storage parameters for the formulations.

Overall, LNPs are the most clinically advanced formulation, as both approved mRNA vaccines are formulated in LNPs, but CNEs and polymeric formulations may be alternatives for future formulations with enhanced stability and efficacy.

## 5. Manufacturing

### 5.1. Production of Self-Amplifying mRNA

saRNA is produced in vitro using an enzymatic transcription in a similar process to the production of conventional shorter mRNA, although the reaction conditions need to be optimized to increase yields for this longer mRNA. The process for the synthesis of in vitro transcribed RNAs was established in the 1990 s [128], predominantly using phage RNA polymerases, and is now a robust and well-established for the large-scale production of synthetic RNA [129]. The production method avoids complex manufacturing and safety issues associated with cell culture production of live viral vaccines, recombinant subunit proteins, and viral vectors (Figure 7). The enzymatic reaction is catalyzed by a phage RNA polymerase, and commercial in vitro transcription kits that produce milligram quantities of RNA for research purposes have been available for several years [67]. Pharmaceutical grade mRNA is currently offered as a contract development and manufacturing organization (CDMO) service by several companies: TriLink (www.trilinkbiotech.com), Aldevron (www.aldevron.com), Eurogentec (www.eurogentec.com), Biomay (www.biomay.com), Creative Biolabs (www.creative-biolabs.com) and several more will enable capacity in the near future. There are no publications describing the large-scale manufacture of saRNA, but Figure 8 describes the unit operations that would be found in a typical cell-free RNA production process [67]. Capped mRNA is produced enzymatically in a bioreactor and the DNA template is digested. DNA fragments, transcription enzymes, reagents, and byproducts are removed using chromatographic purification followed by tangential flow filtration (TFF). During TFF, due to the large size of the saRNA, lower molecular weight species are removed if the appropriate molecular weight cut-off membrane is selected, and the RNA can diafiltered into the appropriate buffer and adjusted to the required concentration. RNA is then sterile filtered and stored in bulk ready for further downstream processing and formulation.

In addition to the polymerase enzyme, in vitro transcription reactions typically includes: A linearized DNA template with a promoter sequence (~23 bases) that has a high binding affinity for its respective polymerase; ribonucleotide triphosphates (rNTPs) for the four required bases (adenine, cytosine, guanine, and uracil); a ribonuclease inhibitor to inactivate any contaminating RNase; a pyrophosphatase to degrade pyrophosphate, which will inhibit transcription; MgCl_2_, which supplies Mg^2+^ as a co-factor for the polymerase; and a pH buffer, which also contains an antioxidant and a polyamine at the optimal concentrations [144,145]. If co-transcriptional capping is utilized, the addition of a cap analogue as an initiator of transcription is required.

The recombinant plasmid is propagated in *Escherichia coli*, linearized using a unique restriction site downstream of the transcription cassette’s 3′ end, and isolated and purified using standard molecular biology techniques. During the in vitro transcription reaction, the bacteriophage polymerase binds the promoter sequence to initiate transcription, and the enzyme then moves along the template towards its 5′ end, elongating the RNA transcript as it travels. Termination of transcription occurs when the enzyme runs off the end of the template (run-off transcription). The poly(A) tail can be encoded into the DNA template, or, alternatively, it can be added enzymatically post-transcription [146]. When the in vitro transcription reaction is complete, the DNA template is fragmented with a DNase, and RNA is recovered using several methods, including precipitation or chromatography. The quality and quantity of RNA produced in an in vitro transcription reaction depends upon a number of factors, including RNA transcript size, template concentration, reaction time and temperature, Mg^2+^ concentration, and NTP concentration [147]. Typically, the conditions require some optimization for each type of construct being produced.

While there is no published data on a large-scale production process for saRNA, the following sections on capping, purification, immunostimulatory by-products, and stability highlight areas that should be consider during process development.

#### 5.1.1. Capping Strategies for saRNA

The in vitro transcribed (IVT) mRNA can be capped either by post-transcriptional modification using capping enzymes [148,149] based on the recombinant vaccinia virus, or by the addition of a cap analog during in vitro transcription [67,148]. Enzymatic capping is more complex but provides much higher yields; capping efficiency is nearly 100% efficient and all capped structures are added in the proper orientation [148]. Enzymatic capping is being used for large-scale and laboratory production, and cap 0 and cap 1 structures can be produced [67]. Co-transcriptional capping with a cap analog is another approach to prepare the IVT mRNA, where a cap analog is provided in excess in the transcription reaction. This process is much simpler compared to the enzymatic capping reaction, but the overall yields tend to be lower and various cap structures can be incorporated with more diverse designs [150,151,152]. The historical issue with the pseudo-symmetrical cap have now been circumvented with anti-reverse cap analogues (ARCAs) [152], which results in a cap 0 structure on approximately 70% of the transcripts and 30% with a 5′ triphosphate. To increase capping efficiency, trimer analogues such as CleanCap [129] have been introduced, which incorporate a cap 1 structure. For saRNA applications, there have not been any published studies comparing the potency of the different capping strategies.

#### 5.1.2. Purification Strategies for saRNA

mRNA has a negatively charged phosphodiester backbone, and many of the purification techniques used for pDNA could potentially be adapted to the purification of this molecule. DNA purification techniques include: Size-exclusion chromatography (SEC), reversed-phase chromatography (RPC), anion-exchange chromatography (AIEX), hydrophobic interaction (HIC), and thiophilic adsorption chromatography (TOC) [153]. For routine pre-clinical work and in vivo immunization studies, RNA can be precipitated. The polar nature of the negatively charged backbone makes RNA highly soluble in water and several cations (lithium chloride is the most widely used) in combination with ice-cold ethanol as a co-solvent can neutralize the backbone charges and decrease solubility to precipitate the RNA out of solution [154]. However, implementing such a process for GMP production would be extremely challenging. Self-amplifying mRNA with sizes in the order of 10,000 bases (MW ~3MDa), has additional challenges over smaller conventional mRNAs and no commercially viable scalable process has been disclosed to date, although likely rely on strategies such as tangential flow filtration (TFF). Review articles on RNA purification [67,155,156] indicate that several techniques could be potentially be utilized and these include: Ion exchange (IE), affinity (AC) and SEC. Thus, there remains a need for improved RNA purification methods for saRNA, that will enable cost and time efficient purification at an industrial scale with high yield and pharmaceutical grade purity, while retaining the stability, biological potency and functionality of the RNA. Large-scale chromatographic purification of saRNA is complex and is an active area of research for many companies and academic institutions.

#### 5.1.3. Immunostimulatory IVT Reaction By-Products

Theoretically the capping strategy could have a positive or negative influence of the potency of the vaccine, since uncapped RNA and different cap structures are known to trigger an antiviral responses [157]. Mechanism of action studies with saRNA vaccines have shown this could potentially lead to reduced potency [88], but there is no published data exploring how the capping strategy could influence the potency of a saRNA vaccine.

The IVT reaction is known to produce by-products that are immune-stimulatory in the form of double stranded RNA (dsRNA). Recent studies have identified two main types of byproducts in the IVT reaction that result in the formation of dsRNA molecules [158,159]. The first is formed by 3′-extension of the run-off products annealing to complementary sequences in the body of the run-off transcript either in *cis* (by folding back on the same RNA molecule) or *trans* (annealing to a second RNA molecule) to form extended duplexes [160]. The second type of dsRNA molecules is formed by the hybridization of an antisense RNA molecule to the run-off transcript [129], produced by promoter-independent transcriptional initiation. dsRNA has been removed from conventional base-modified mRNA using ion pair reversed-phase HPLC [161], but this method is not scalable, the acetonitrile eluent is very toxic, and there is no evidence it would work for saRNA. A better approach, although not tested with saRNA, would be to utilize the selective binding of dsRNA to cellulose in an ethanol-containing buffer [161]. Alternatively, as described recently, mRNA can be produced by combining high-temperature IVT with template-encoded poly(A) tailing [162]. This process reduced the formation of both kinds of dsRNA by-products, generating functional mRNAs with reduced immunogenicity. It should be noted that neither of these techniques were used with larger saRNA. Theoretically, the presence of dsRNA could have a positive or negative influence of the potency of the vaccine since dsRNA is known to trigger antiviral responses [153]. Mechanism of action studies with saRNA vaccines have shown this could potentially lead to reduced potency [154], but there are no published data exploring how the presence of dsRNA could influence the potency of an saRNA vaccine.

#### 5.1.4. Stability of mRNA

There are considerable differences in stability between DNA and RNA. With over 20 years of extensive research and development of pDNA vaccines, a rationally designed liquid formulation that is stable for 1 year at 30 °C has been developed [163]. This degree of stability is unlikely in mRNA vaccines, because RNA contains a 2′hydroxyl group on the ribose, which is hydrolytically much less stable than the deoxyribose. Theoretical calculations for the 5 °C stability for a “naked” 4000 nucleotide mRNA in bulk solution (PBS, pH 7.4, no Mg^2+^) estimate a half-life of 900 days [164]. However, a rise in temperature to 37 °C is predicted to lead to a reduced half-life of 5.4 days. Longer self-amplifying mRNA (12 kB) was calculated to have exacerbated hydrolysis (3-fold higher), with an expected half-life of 314 days at 5 °C and 2 days at 37 °C.

During production (IVT reaction and downstream processing) the mRNA is subject to high concentrations of Mg^2+^ and elevated temperatures [165], which need to be mitigated to limit hydrolysis. A largely unexplored strategy and theoretical basis to reduce mRNA hydrolysis is to redesign RNAs to form double-stranded regions, which are protected from in-line cleavage and enzymatic degradation, while coding for the same proteins [164].

RNA is more sensitive than DNA to oxidation, alkylation or electrophilic additions which result in hydrolysis of glycosidic bonds [163]. In addition, RNA is prone to enzymatic degradation with three major classes of RNA-degrading enzymes (ribonucleases or RNases): Endonucleases (which cut RNA internally), 5′ exonucleases (which hydrolyze RNA from the 5′ end), and 3′ exonucleases (which degrade RNA from the 3′ end) [163]. Therefore, after production of the saRNA it is generally stored at −80 °C and great care is taken to avoid the introduction of RNases. The optimal pH to store RNA is in the range of pH = 4–5 [4], since RNA is susceptible to alkaline hydrolysis at pH > 6, and acid hydrolysis only occurs at pH < 2.

During delivery after intramuscular vaccination, saRNA is susceptible to hydrolysis due to the presence of high levels of Mg^2+^ ions and a body temperature of 37 °C, and RNase degradation. Encapsulation in an LNP has been shown to limit enzymatic degradation [52], but it should be noted that the saRNA encapsulated in lipid formulations may be subject to increased hydrolysis if the lipid’s cationic headgroups lower the pKa of the ribose 2′ hydroxyl group [164].

### 5.2. Manufacturing Considerations for Formulated mRNA Drug Product

While the manufacturing and production process for the formulated mRNA drug product can differ considerably depending on the type of formulation, a clinically relevant manufacturing process can be generalized into four steps: (1) Formulation, which involves one or more mixing steps, (2) downstream processing and purification, (3) sterile filtration through a 0.2 µm filter, (4) fill and finish. Presently, little information has been published regarding the specific manufacturing processes utilized for saRNA vaccine candidates currently in clinical trials. Hence, the preclinical processes for each formulation type will be generalized from methods published in the literature for lipid nanoparticles and nanoemulsions. To focus on potential clinical production, scalable continuous flow process steps are favored over fixed-volume processes.

#### 5.2.1. Production of Lipid Nanoparticles

Anderluzzi et al. demonstrated the versatility of the NanoAssemblr microfluidic mixers to produce a variety of saRNA formulations such as liposomes, solid lipid nanoparticles and polymer nanoparticles by continuous flow solvent/antisolvent precipitation [86]. Lou et al. also demonstrated the use of the same platform for saRNA formulations with ionizable LNPs [87]. The technology has been established for producing other RNA-LNP formulations [53,166,167,168,169] including base-modified mRNA vaccines [27,170,171]. The process involved rapid advective mixing of a water-miscible organic solvent containing dissolved lipids or polymers with an aqueous phase containing dissolved saRNA at optimized flow rates and organic/aqueous flow rate ratios to control the precipitation conditions. When using ionizable cationic lipids, the aqueous phase containing dissolved saRNA is buffered at pH 4–below the pKa of the ionizable lipid. Other in-line mixing methods have also been employed for saRNA-LNP formulations including alternative microfluidic architectures [71] and in-line macro-mixing in a T-tube [18,52,66,75,88]. To remove solvent and adjust the final concentration, tangential flow filtration has been employed as a high-throughput method [18,88].

#### 5.2.2. Product of Nanoemulsions

Cationic nanoemulsions employ two immiscible phases and thus required a different method for production. The processes described in the literature generally involved dissolving the cationic lipid and a hydrophobic surfactant in squalene. The resulting oil phase is mixed with a mildly acidic buffer containing a hydrophilic surfactant to create a primary emulsion. The primary emulsion is then repeatedly passed through a high-pressure homogenizer to obtain a more homogeneous nanoemulsion. The resulting emulsion is complexed with saRNA by mixing and incubating at 4 °C for 30 to 120 min [68,74,86]. The formulation is then sterilized by passage through a 0.2 µm filter [91]. High-pressure homogenization has been established for large scale production of lipid-based colloids for drug delivery [172,173,174,175,176].

Overall, large-scale production of RNA formulations has been streamlined by co-transcriptional capping and chromatographic purification to remove double-stranded RNA, and microfluidic systems that enable reproducible batches of particles.

## 6. Future Outlook

While historical (pre-2015) preclinical studies of saRNA vaccines were predominantly focused on viral replicon particles and cancer applications, the field has more recently shifted to applications in viral infectious diseases, although a few studies have also explored prevention of parasitic and bacterial infections. The investigation of saRNA for passive immunization by encoding a monoclonal antibody is also a highly promising application that warrants further development. The clinical trials for rabies and SARS-CoV-2 are an exciting opportunity for the field of saRNA vaccines, and will no doubt be informative as to the characteristics of the immune response, required dose, duration of immunity, and required regimen. The field is also starting to consider methods to modulate the innate response to saRNA, which will no doubt be imperative to the clinical success of these vaccines, so that the lesson of DNA vaccine clinical trials are not forgotten [177]. One strategy that may facilitate efficacious saRNA vaccines is utilizing evaluation models, such as skin explants that have human immune cells and innate sensing, in order to optimize molecular and delivery components. While the SARS-CoV-2 global pandemic has been detrimental to economies and health, it’s provided a valuable opportunity to test saRNA vaccines in the clinical that otherwise might have taken decades. Given the short timespan required to design and test new saRNA vaccines (reportedly as little as 14 days in the case of Imperial College London) [178], it is clear that this platform is particularly well-suited to outbreaks, and also possibly seasonal vaccines, such as influenza. The rapid and easy manufacture of saRNA vaccines may also pave the way for a distributed manufacturing model where vaccines are produced locally in order to minimize logistical and cold-chain issues that could hinder widespread distribution of a vaccine. While immense progress was made in RNA vaccine technology in 2020 [34], the main limitations are now the stability, which requires storage at <80 °C for most RNA formulations [179], and minimizing the required dose in order to reduce associated side effects [180]. Overall, saRNA vaccines have made monumental strides in the past five years, and the next five years will be telling as to the clinical utility and success of this promising vaccine platform.

## Figures and Tables

**Figure 1 vaccines-09-00097-f001:**
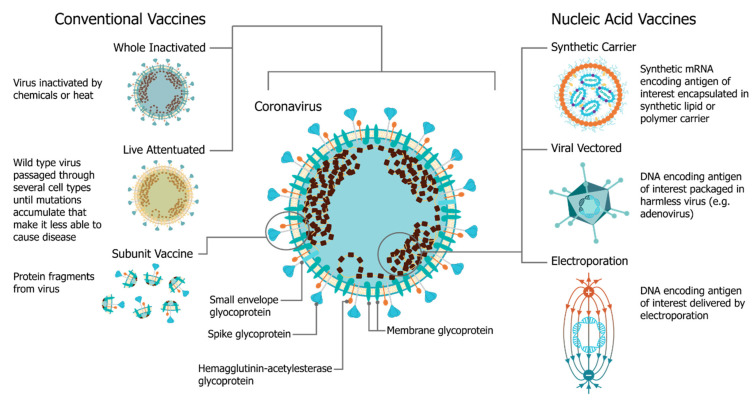
A comparison of vaccine platforms including vaccines derived from the virus itself and are formulated as a part or whole modified version of the virus (**left**) and nucleic acid vaccines, such as self-amplifying RNA vaccines (**right**). Nucleic acid vaccines are derived from knowledge of the viral genome, where glycoproteins are encoded into nucleic acids and delivered with either a synthetic carrier such as a lipid nanoparticle or an inert viral delivery system such as adenoviruses. The encoded antigen sequences are then expressed by the host cells.

**Figure 2 vaccines-09-00097-f002:**
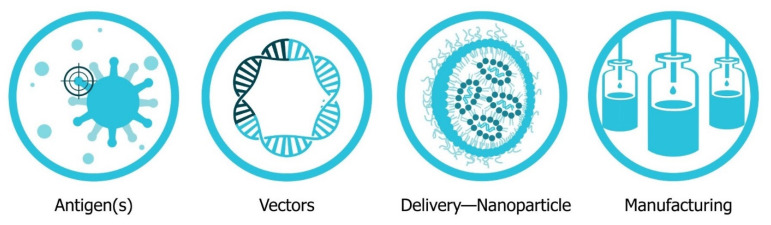
The Four Pillars of successful saRNA vaccine development. The antigens, vectors, delivery and manufacturing each represent modular components that need to be combined to make a successful drug product. Each pillar has its set of design and development considerations and associated technologies that are explored in this review.

**Figure 3 vaccines-09-00097-f003:**
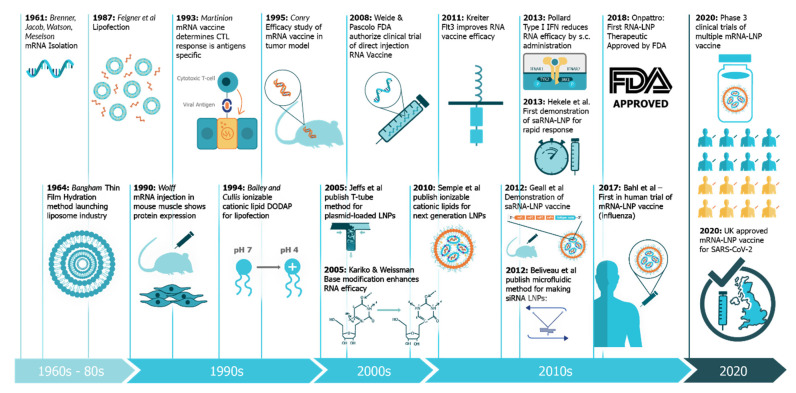
A timeline of innovations that have contributed to the development of saRNA vaccines and associated technologies. These include advances in technologies associated with each of the Four Pillars of a successful saRNA vaccine [27,39,40,41,42,43,44,45,46,47,48,49,50,51,52,53,54,55,56].

**Figure 4 vaccines-09-00097-f004:**
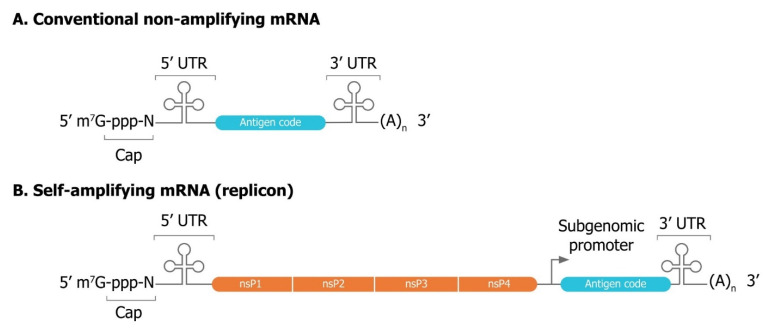
A comparison of mRNA vectors. Both conventional (**A**) and self-amplifying (**B**) mRNAs share basic elements including a cap, 5′ UTR, 3’ UTR, and poly(A) tail of variable length. Self-amplifying RNA (saRNA) also encode four non-structural proteins (nsP1–4) and a subgenomic promoter derived from the genome of the alphavirus. nsP1–4 encode a replicase responsible for amplification of the saRNA that enable lower doses than non-replicating mRNA.

**Figure 5 vaccines-09-00097-f005:**
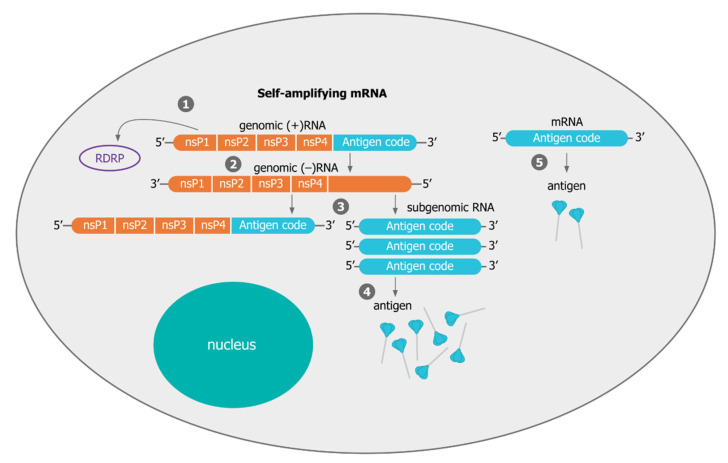
Mechanism of self-amplifying mRNA. (**1**) Following delivery to the cytoplasm, translation of the saRNA produces the non-structural proteins 1–4 (nsP 1–4) that form the (RDRP). (**2**) RDRP is responsible for replication of the saRNA producing copies of the saRNA. Multiple copies of the subgenomic RNA (**3**) are hence produced from each saRNA originally delivered. This leads to translation of many more copies of the antigen (**4**) when compared to a non-amplifying RNA (**5**).

**Figure 6 vaccines-09-00097-f006:**
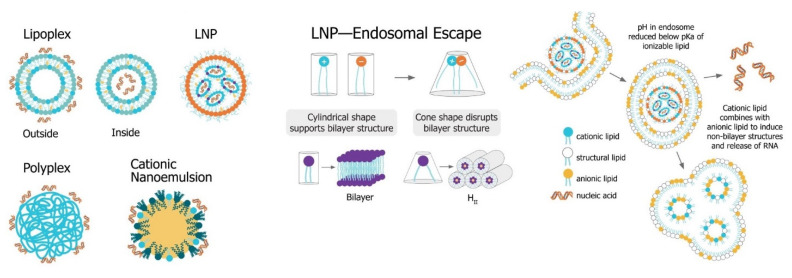
Non-viral saRNA delivery systems. Lipid-, polymer-, and emulsion-based delivery systems all use cationic groups to mediate condensation of the anionic RNA as well as delivery across the cell membrane. LNP systems, which have been found to be the most potent vaccine formulatinos, utilize a pH-sensitive ionizable cationic lipids and are taken up in cells through receptor-mediated endocytosis. In the endosome, the lower pH environment ionizes the cationic lipids, which then interacts electrostatically with anionic lipids in the endosomal membrane. These ion pairs cause a phase transition into a porous hexagonal phase (H_II_) that disrupts the endosome and facilitates release of the RNA into the cytoplasm.

**Figure 7 vaccines-09-00097-f007:**
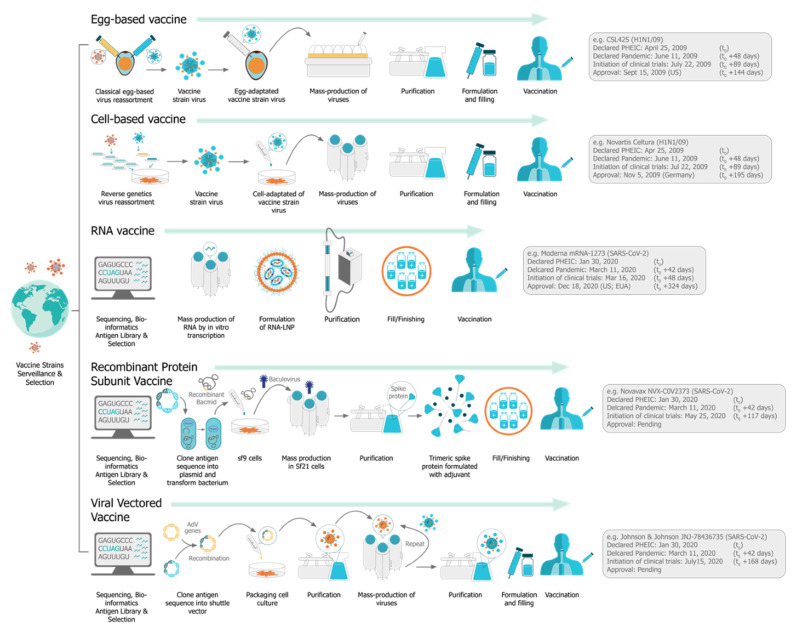
A comparison of vaccine drug product manufacturing processes for egg- and cell-based manufacturing of conventional vaccines, as well as vaccines produced from viral genome sequence information such as the RNA, protein subunit, and viral vectored DNA vaccines against SARS-CoV-2 from Moderna, Novavax, and Johnson & Johnson respectively [130,131,132,133,134,135,136,137,138,139,140,141,142,143]. RNA vaccines offer a cell-free manufacturing process that is responsible for many advantages of the platform, allowing facile and rapid vaccine manufacturing. Moderna’s mRNA vaccine against SARS-CoV-2 (mRNA-1273) began clinical trials just 63 days following the publication of the SARS-CoV-2 genome. * For comparative purposes, we have included historical timelines for the flu pandemic vaccines for egg and cell culture production, but it should be noted that large efficacy trials are not required for these vaccines since they are licensed based on a correlate of protection (hemagglutination inhibition (HI) antibody responses).

**Figure 8 vaccines-09-00097-f008:**
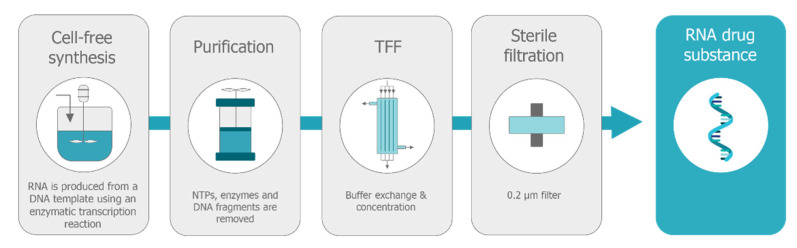
Schematic diagram of the manufacturing process for the RNA drug substance. The process involves a cell-free enzymatic in-vitro transcription reaction followed by purification to remove the DNA template, followed by tangential flow filtration (TFF) for buffer exchange and concentration, followed by sterile filtration through a 0.2 µm filter.

**Table 1 vaccines-09-00097-t001:** Published preclinical and clinical trial data with mRNA COVID-19 vaccines.

Sponsor	Type of mRNA	Delivery System	Preclinical Data	Clinical Data
Moderna	bmRNA	LNP	[9,12]	[7,8]
BioNTech/Pfizer	bmRNA	LNP	[13]	[11,14,15,16,17]
ICL	saRNA	LNP	[18]	
Arcturus	saRNA	LNP	[19]	
CureVac	mRNA	LNP	[20]	

Imperial College London (ICL), conventional non-amplifying messenger ribonucleic acid (mRNA), conventional base-modified non-amplifying mRNA (bmRNA) and self-amplifying messenger RNA (saRNA).

**Table 2 vaccines-09-00097-t002:** Preclinical testing of saRNA vaccines against infectious diseases and cancer since 2015.

Disease Category	Disease Target	Replicon Backbone	Antigen	Delivery Platform	Preclinical Animal Model	Ref.
Infectious Disease	*Chlamydia trachomatis*	VEEV	MOMP	CAF, PEI	Mice	[57]
	Ebola	VEEV	Glycoprotein (EBOV)	Dendrimer	Mice	[59]
	Group A *Streptococci*	VEE-SINV	GAS SLOdm	CNE	Mice	[58]
	Group B *Streptococci*	VEE-SINV	GBS BP-2a	CNE	Mice	[58]
	HCV	VEEV	E1-E2	CNE	Mice	[67]
	HCMV	VEEV	gH/gL	LNP	Mice	[67]
	HIV-1	VEE-SINV	TV1 Env gp140	CNE	NHP	[68]
		SFV	Gag/Pol Mosaic	PEI	Mice	[69]
		VEEV	eOD-GT8 gp120	LNP	Mice	[70]
		VEEV	Env gp140	Lipoplex	Mice	[71]
		SFV	HIV-1C Env, Gag, PolRT	Naked	Mice	[72]
	Malaria	VEE-SINV	PMIF	CNE	Mice	[73]
	Influenza	VEEV	HA (H1N1, A/WSN/33)	Dendrimer	Mice	[59]
		VEE-SINV	HA (H1N1, A/Cal/7/09)	CNE	Mice, Ferrets	[74]
		VEE-SINV	NP (H1N1, A/PR/34/07)	LNP	Mice	[75]
		VEE-SINV	NP, M1 or NP+M1 (H1N1, A/PR/8/34)	LNP	Mice	[66]
		CSFV	HA, NP (H5N1/Yamaguchi/2004)	PEI with CPP	Mice, Pigs	[76]
		CSFV	NP (H3N2, Brisbane 2007)	Cationic lipid	Mice	[77]
		n.s.	HA (H1N1, A/PR/8, A/Cal/7/09)	PEI	Mice	[78]
		CSFV	HA, NP (H5N1/Yamaguchi/2004)	PEI	Mice	[79]
		VEEV	HA (A/PR/8/34)	LPPs	Mice	[80]
		n.s.	HA (H1N1, A/Cal/7/09)	MLNPs	Mice	[81]
		SFV taRNA	HA (H1N1, A/Cal/7/09)	Naked	Mice	[82]
		VEEV	HA (H1N1, A/Cal/7/09)	pABOL	Mice	[83]
		VEE-SINV	NP, GMCSF	CNE	Mice	[84]
	Rabies	VEE-SINV	Glycoprotein G	CNE	Rats	[85]
		VEE-SINV	Glycoprotein G	PNPs, Liposomes, CNE	Mice	[86]
		VEE-SINV	Glycoprotein G	LNP, CNE	Mice	[87]
		VEEV	Glycoprotein G	LNP, CNE	Mice	[67]
	Respiratory syncytial virus	VEE-SINV	Glycoprotein F	LNP	Mice	[88]
	SARS-CoV-2	VEEV	Pre-fusion stabilized spike protein	LNP	Mice	[18]
		VEEV	Spike protein	LION emulsion	Mice, NHP	[89]
		VEEV	Pre-fusion spike protein	LNP	Mice	[62]
		VEEV	Spike protein	LNP	Mice	[19]
	*Toxoplasma gondii*	VEEV	GRA6, ROP2a, ROP18, SAG1, SAG2A, AMA1	Dendrimer	Mice	[59]
		SFV	NTPase-II	LNP	Mice	[60]
	VEEV	VEEV	E3-E2-6K-E1	CNE	Mice	[65]
	Zika	VEEV	prM-E	Dendrimer	Mice	[90]
		VEEV	prM-E	NLC	Mice, Guinea pigs	[91]
		VEEV	prM-E	Naked	Mice	[92]
		VEEV	ZIKV-117 Ab	NLC	Mice	[63]
		n.s.	prM-E	CNE	Mice, NHPs	[93]
		VEEV	NS3, prM-E	LNP	Mice	[94]
Cancer	Melanoma	VEEV	IL-12	LNP	Mice	[61]
		VEEV	IL-2	LNP	Mice	[62]
	Colon carcinoma	VEEV	IL-12	LNP	Mice	[61]

n.s. = Not specified, Antibody (Ab), Apical membrane antigen 1 (AMA1), Cationic adjuvant formulation (CAF), Cationic nanoemulsion (CNE), Cell penetrating peptides (CPP), Classical swine fever virus (CSFV), Dense granule protein 6 (GRA6), E1-E2 glycoproteins of hepatitis C virus (E1-E2), GAS Streptolysin-O (SLOdm), GBS pilus 2a backbone protein (BP-2a), gH and gL glycoproteins of human cytomegalovirus (gH/gL), Granular-macrophage colony-stimulating factor (GM-CSF), Group specific antigen (Gag), Envelope protein (Env), Hemagglutinin (HA), human cytomegalovirus (HCMV), hepatitis C virus (HCV), Interleukin-2 (IL-2), Interleukin-12 (IL-12), Lipid inorganic nanoparticle (LION) emulsion, Lipid nanoparticles (LNPs), Major Outer Membrane Protein (MOMP), Mannosylated lipid nanoparticles (MLNPs), Membrane protein 1 (M1), Nanostructured lipid carrier (NLC), Nonhuman primate (NHP), Nucleoprotein (NP), Nucleoside Triphosphate Hydrolase-II (NTPase-II), poly(CBA-*co*-4-amino-1-butanol (pABOL), Poly(ethylene imine) (PEI), *Plasmodium* macrophage migration inhibitory factor (PMIF), Polymerase protein (Pol), Polymeric nanoparticles (PNPs), Pre-membrane and envelope protein (prM-E), Reverse transcriptase (RT), Rhoptry protein 2A (ROP2A), Rhoptry protein 18 (ROP18), Self-amplifying RNA (saRNA), Self-amplifying and replicating RNA (STARR™), Semliki Forest Virus (SFV), Severe acute respiratory syndrome coronavirus 2 (SARS-CoV-2), Surface antigen 1 (SAG1), Surface antigen 2A (SAG2A),Venezuelan equine encephalitis virus (VEEV), *trans*-amplifying RNA (taRNA), Venezuelan equine encephalitis and Sindbis virus replicon chimera (VEE-SINV).

**Table 3 vaccines-09-00097-t003:** Clinical trials of saRNA vaccines since 2015.

Disease Target	Institution	Vaccine Components (Route of Administration)	Target	Clinical Trial Number (Phase)	Status
Rabies	GlaxoSmithKline	VEE-SINV saRNA with CNE (IM)	Rabies glycoprotein G	NCT04062669 (I)	Ongoing, recruiting
SARS-CoV-2	Arcturus Therapeutics	STARR™ (VEEV) saRNA with LUNAR^®^ LNP (IM)	Pre-fusion stabilized spike protein of SARS-CoV-2	NCT04480957 (I)	Ongoing, recruiting
	HDT Bio Corp.	VEEV saRNA with LION emulsion (IM)	Spike protein of SARS-CoV-2	-	Pre-recruiting
	Imperial College London	VEEV saRNA with LNPs (IM)	Pre-fusion stabilized spike protein of SARS-CoV-2	ISRCTN17072692 (II)	Ongoing, recruiting
	Imperial College London, University of Oxford	VEEV saRNA with LNPs OR ChAdOx (IN)	Pre-fusion stabilized spike protein of SARS-CoV-2	-	Pre-recruiting
Non-Small Cell Lung Cancer, Colorectal Cancer, Gastroesophageal Adenocarcinoma, Urothelial Carcinoma	Gritstone Oncology, Inc.	GRT-C901, GRT-R902	Personalized neoantigens	NCT03639714 (I/II)	Recruiting
Non-Small Cell Lung Cancer, Colorectal Cancer, Pancreatic Cancer, Solid Tumor, Shared Neoantigen-Positive Solid Tumors	Gritstone Oncology, Inc.	GRT-C903 GRT-R904	Personalized neoantigens	NCT03953235 (I/II)	Recruiting

Cationic nanoemulsion (CNE), Chimpanzee adenovirus-vectored vaccine (ChAdOx), Intranasal (IN), Intramuscular (IM), Lipid nanoparticles (LNPs), Lipid-enabled and Unlocked Nucleomonomer Agent (LUNAR^®^), Self-amplifying RNA (saRNA), Self-amplifying and replicating RNA (STARR™), Severe acute respiratory syndrome coronavirus 2 (SARS-CoV-2), Venezuelan equine encephalitis virus (VEEV), Venezuelan equine encephalitis and Sindbis virus replicon chimera (VEE-SINV).

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
