# Peer review of "An Update on Self-Amplifying mRNA Vaccine Development"

_vaccines, 2021, doi:10.3390/vaccines9020097_

Round 1

Reviewer 1 Report

The manuscript written by Geall et al is a recent review on an update on self-amplifying mRNA vaccine development, I recommend publication of the paper after some minor revisions on "Vaccines".

The authors start from the brief introduction of the four pillars of saRNA vaccines, focused on antigen design, vector design, delivery systems and manufacturing. 

Figure 2 & 3 needs to be improved.

The authors may also want to add more in detail about

(1) plasmid DNA vaccines

(2) limitations of mRNA vaccine development and current methods used to overcome the limitations. 

(3) how nucleic acid vaccines can be used as a versatile tool for human immunization.

(4) mechanism of immune response induced by mRNA Vaccines

(5) delivery route and formulation of mRNA Vaccines

Author Response

Reviewer 1:

The manuscript written by Geallet al is a recent review on an update on self-amplifying mRNA vaccine development, I recommend publication of the paper after some minor revisions on "Vaccines".The authors start from the brief introduction of the four pillars of saRNA vaccines, focused onantigen design, vector design, delivery systems and manufacturing.

Thank you for the thorough review, see a point-by-point response to your comments below in bold.Change to the manuscript are highlighted in orange.

Figure 2 & 3 needs to be improved.We have improved

Figures 2, 3, 5, 7and 8to include more detail and improved graphics.

The authors may also want to add more in detail about

(1) plasmid DNA vaccines

(2) limitations of mRNA vaccine development and current methods used to overcome the limitations.

(3) how nucleic acid vaccines can be used as a versatile tool for human immunization.

(4) mechanism of immune response induced by mRNA Vaccines

(5) delivery route and formulation of mRNA Vaccines

As the aim of our review was to provide an update on the progress of saRNA vaccines in the last 5 years, we did not include specific sections on each of these topics. However, we agree that these are important aspects of mRNA vaccines and have included references to other reviews that cover these topics in the introduction:

Pertinent reviews on plasmid DNA and non-replicating messenger RNA vaccines can be found at the following references, which provide insight into the mechanism of immune response and effects of route of delivery.[34-37]

We have included more detail on the versatility of saRNA vaccines as a tool for human immunization in the introduction:

This is especially advantageous as a vaccine platform as mRNA vaccines can be produced for any pathogen with a known protein target.

Finally, we have included limitations of saRNA vaccines in the ‘Future Outlook’ section:

While immense progress was made in RNA vaccine technology in 2020,[34]the main limitations are now the stability, which requires storage at < 80 °C for most RNA formulations,[166]and minimizing the required dose in order to reduce associated side effects.[167]

Reviewer 2 Report

The manuscript by Blakney et al. overviews four major steps of saRNA based development including antigen design, vector development, available delivery systems, and production. The review is logically organized and well written. Display items are very informative and self-explanatory. While this review is not very comprehensive in each aspect of saRNA vaccine development it still provides a substantial overview of the emerging field. However, I think the weakest point of the review is the lack of a critical point of view from the authors on the subject. For example, the author listed and described three different designs of vectors (P8-11) in three independent subsections. I got two major questions after reading this part. First, what are the advantages and drawbacks of each design, and second how these designs compared to each other. In addition, it will be helpful to include perspectives on the vector design. Are there new vector designs with high potential? Or is one of these designs more popular than others and will dominate in the future? I have similar thoughts regarding other sections of the manuscript. I hope the authors can address my comments and make this review a bit more critical.

The minor points. I encountered some typos and minor mistakes throughout the text. Figure 7, Egg-based and cell-based vaccines do not contain a timeline for production, while all others have time indicated, please be consistent.

Author Response

Reviewer 2:

The manuscript by Blakney et al. overviews four major steps of saRNA based development including antigen design, vector development, available delivery systems, and production. The review is logically organized and well written. Display items are very informative and self-explanatory. While this review is not very comprehensive in each aspect of saRNA vaccine development it still provides a substantial overview of the emerging field.

Thank you for the thorough review, see a point-by-point response to your comments below in bold. Change to the manuscript are highlighted in orange.

However, I think the weakest point of the review is the lack of a critical point of view from the authors on the subject. For example, the author listed and described three different designs of vectors (P8-11) in three independent subsections. I got two major questions after reading this part. First, what are the advantages and drawbacks of each design, and second how these designs compared to each other. In addition, it will be helpful to include perspectives on the vector design. Are there new vector designs with high potential? Or is one of these designs more popular than others and will dominate in the future? I have similar thoughts regarding other sections of the manuscript. I hope the authors can address my comments and make this review a bit more critical.

Thank you for this feedback; we have added the following constructive criticisms to each the vector design, delivery and manufacturing sections:

Overall, VEEV and SINV vector designs have been shown to have the most promising vaccine immunogenicity, and are being improved by next-generation designs such as trans-amplifying replicons and incorporation of interferon inhibiting factors directly into the vector backbone.

Overall, LNPs are the most clinically advanced formulation, as both approved mRNA vaccines are formulated in LNPs, but CNEs and polymeric formulations may be alternatives for futureformulations with enhanced stability and efficacy.

Overall, large-scale production of RNA formulations has been streamlined by co-transcriptional capping and chromatographic purification to remove double-stranded RNA, and microfluidic systems that enable reproducible batches of particles.

The minor points. I encountered some typos and minor mistakes throughout the text.

We have fixed the typos and minor mistakes in the text, which are highlighted in orange throughout.

Figure 7, Egg-based and cell-based vaccines do not contain a timeline for production, while all others have time indicated, please be consistent.

Thank you, we have included a timeline for egg-and cell-based vaccines for consistency.
